# Non-Coding RNAs Participate in the Regulation of *CRY-DASH* in the Growth and Early Development of *Saccharina japonica* (Laminariales, Phaeophyceae)

**DOI:** 10.3390/ijms21010309

**Published:** 2020-01-02

**Authors:** Xiaoqi Yang, Lu Li, Xiuliang Wang, Jianting Yao, Delin Duan

**Affiliations:** 1Key Laboratory of Experimental Marine Biology, Center for Ocean Mega-Science, Institute of Oceanology, Chinese Academy of Sciences, Qingdao 266071, China; yangxiaoqi@qdio.ac.cn (X.Y.); lilu14@mails.ucas.ac.cn (L.L.); xlwang@qdio.ac.cn (X.W.); yaojianting@ms.qdio.ac.cn (J.Y.); 2Laboratory for Marine Biology and Biotechnology, Qingdao National Laboratory for Marine Science and Technology, Qingdao 266071, China; 3University of Chinese Academy of Sciences, Beijing 100049, China

**Keywords:** blue light, circadian rhythm, CRY-DASH, non-coding RNA, growth, *Saccharina japonica*

## Abstract

CRY-DASH, a new cryptochrome blue light receptor, can repair damaged DNA and regulate secondary metabolism and development of fungus. However, its role in regulation during the growth of *Saccharina japonica* is still unclear. After cloning the full-length of *CRY-DASH* from *S. japonica* (*sjCRY-DASH*), we deduced that its open reading frame was 1779 bp long and encoded 592 amino acids. *sjCRY-DASH* transcription was rapidly upregulated within 30 min in response to blue light and exhibited 24 h periodicity with different photoperiods. Moreover, *sjCRY-DASH* maintained the same periodicity in suitable growth temperature, suggesting a close relationship between this periodicity and circadian rhythm regulation. *Novel-m3234-5p*, which was targeted to *sjCRY-DASH*, decreased with increasing *sjCRY-DASH* transcription, acting as a negative modulator of *sjCRY-DASH*. Six long non-coding RNAs classified as long intergenic non-coding RNAs (lincRNAs) exhibited co-expression with *sjCRY-DASH*. A miRNA *sjCRY DASH* lincRNA network was consequently identified. By predicting the endogenous competing mRNAs of *novel*-*m3234*-*5p*, we found that *sjCRY-DASH* indirectly participated in the regulation of DNA damage repair, protein synthesis and processing, and actin transport. In conclusion, our results revealed that non-coding RNAs participate in the regulation of *sjCRY-DASH,* which played vital roles in the growth and early development of *S. japonica*.

## 1. Introduction

Members of the cryptochrome/photolyase family, an ancient group of ubiquitously distributed UV-A/blue light (BL) sensitive proteins, exhibit similar structures but very diverse functions [1,2]. Photolyase is a flavin photoenzyme that can repair damaged DNA by transferring electrons from bound flavin to damaged pyrimidine dimers [3,4]. Cryptochromes that exhibit significant homology to photolyases play major roles in regulating plant photomorphogenesis [5], the circadian clock [6], and the growth and development of seedlings [7].

Phylogenetically, there are three types of cryptochromes: Animal cryptochrome, plant cryptochrome, and cryptochrome DASH (CRY-DASH) [8]. Cryptochromes from both plants and animals function in the photoentrainment of circadian rhythms [5], but plant cryptochromes primarily mediate BL stimulation, which is an input pathway for the circadian clock. CRY-DASH, a new member of the cryptochrome, is widely distributed from bacteria to vertebrates, indicating that these cryptochromes are most likely present in the last common ancestor of eukaryotes [9,10]. CRY-DASH is characterized by an N-terminal photolyase related domain and a C-terminal domain of varying length [11]. Structural analysis reveals that CRY-DASH targets both chloroplasts and mitochondria by transient peptide sequences [12].

CRY-DASH can repair cyclobutane pyrimidine dimer lesions in single stranded DNA and in loop structures of double stranded DNA with the same efficiency [8]. Additionally, CRY-DASHs are involved in light regulation during the development and secondary metabolism of fungi [12,13,14]. In *Synechocystis*, a *CRY-DASH* that is sensitive to BL participates in the assembly of the photosystem II reaction center complex [15]. Many *CRY-DASH* genes transcribed in the green algae *Ostreococcus tauri* are active in the light/dark cycle [11,16]. CRY-DASH genes transcribed in the dinoflagellate *Karenia brevis* are active in the cell cycle [17]. A novel cryptochrome that exhibits the typical characteristics of *CRY-DASH* can regulate the expression of light harvesting proteins in the diatom *Phaeodactylum tricornutum* [18]. Therefore, the function of these characterized CRY-DASHs seems to be species-specific and associated with the diverse regulation of growth and development. Nevertheless, little is known about CRY-DASH in macroalgae, which exhibit intricate mechanisms to adjust their physiology and growth to cope with physically stressful habitats.

*Saccharina japonica* (kelp), one of the most economically important seaweeds, inhabits littoral zones where BL is predominant [19]. Kelp has thus evolved a complex response mechanism to adapt to the BL environment. Transcriptome comparative analysis has identified seven cryptochromes that are sensitive to BL, and CRY-DASH is one of the candidate genes. However, the regulatory mechanism controlling CRY-DASH has not been verified in kelp. Non-coding RNAs have recently been accepted to be involved in the regulation of light signals [20,21]. MicroRNAs (miRNAs) that evolved independently have been widely identified in brown algae by high throughput sequencing [20,22,23]. MiRNAs exhibit high divergence and rapid evolution between *Ectocarpus* sp. and *S. japonica*. The preference for uracil in the first residue of miRNA which is linked to sorting by Argonaute proteins is not as marked in *S. japonica* as it is in *Ectocarpus* [20,23,24]. In addition, long non-coding RNAs (lncRNAs) in brown algae contain both conserved and species-specific domains and might have different functions [25]. The existence of lncRNAs in brown algae reflects their variation with ongoing global changes [26]. Nevertheless, the mechanisms of lncRNA action need to be explored, especially the mechanisms of non-coding RNAs.

In this study, we characterized the molecular structure of CRY-DASH and predicted its regulatory mechanism and biological function in *S. japonica*. Its relationship with non-coding RNAs allowed us to discover the molecular mechanism of *CRY-DASH* in the growth and early development of kelp.

## 2. Results

### 2.1. Characterization of CRY-DASH in S. japonica

The complete cDNA sequence of s*jCRY-DASH* was obtained through amplification with 3′-rapid amplification of cDNA ends (RACE) and 5′-RACE PCR. *SjCRY-DASH* was characterized by a 122-bp 5′-untranslated region (UTR), a 1779-bp of open reading frame (ORF) that encoded 592 amino acids, and a 459-bp region of the 3′-UTR (Figure 1). The calculated molecular mass was estimated as 63.827 kDa by Compute isoelectric point (pI)/molecular weight (MW) (http://web.expasy.org/compute_pi/). To characterize the expression of *sjCRY-DASH*, the full sequence of *sjCRY-DASH* was recombined in the *pCold I* plasmid and then transformed into *Escherichia coli* for heterologous expression. After the supernatant was run through a His affinity column, one distinct approximately band at approximately 66 kDa was visualized with sodium dodecyl sulfate polyacrylamide gel electrophoresis (SDS-PAGE) (Figure 2A), which was consistent with the predicted MW of the recombinant protein. A positive fraction for anti-His antibody exhibited a single band at MW of the target protein (Figure 2B). In addition, phylogenetic analysis showed that sjCRY-DASH was more distantly related to the CRY-DASHs of other photosynthetic organisms than to those of vertebrates (Appendix A); however, sjCRY-DASH was clustered into the CRY-DASH subfamily of the cryptochrome/photolyase family (Appendix A).

### 2.2. sjCRY-DASH Transcription is Rapidly Upregulated by Blue Light

Both BL and white light (WL) upregulated transcription, but there was no significant change with red light (RL) (Tukey’s test, *p* = 0.09; Figure 3). *sjCRY-DASH* transcription under BL increased by 3.68-fold, 5.89-fold, 6.10-fold, 2.02-fold, and 3.94-fold at 10, 30, 60, 180, and 300 min, respectively. Under WL, it increased by 1.58-fold, 2.12-fold, 6.54-fold, 4.84-fold, and 4.36-fold. Although the transcriptional levels were similar under BL and WL at 1 h, they were significantly higher in BL than in WL at 10 min and 30 min (Tukey’s test, *p* < 0.05 for all comparisons). The rapid response of *sjCRY*-*DASH* to BL indicated that *sjCRY-DASH* was closely associated with BL.

### 2.3. Photoperiod Affects the Circadian Oscillation of sjCRY-DASH

The response of transcript levels of s*jCRY-DASH* in response to different photoperiods was detected at 4 h intervals. With a light/dark (L/D) 16:8 illumination period, the transcript level of *sjCRY-DASH* increased from Zeitgeber Time 2 (ZT2) to ZT6, rapidly decreased from ZT6 to ZT14, then increased at ZT18 and decreased from ZT18 to ZT22 (Figure 4A). In the 16:8 group, the number of transcripts peaked at ZT6 and ZT30. Moreover, the 24 h variation in the maximum transcription of *sjCRY-DASH* was coincident at L/D 12:12 and L/D 8:16, which indicated that *sjCRY-DASH* exhibited 24 h periodicity. For L/D 12:12, transcription increased by 12.55-fold at ZT6 compared to ZT2, while at ZT14, transcription decreased by 0.10-fold (Figure 4B), implying that circadian oscillation was more noticeable in the L:D 12:12 group than in the L/D 16:8 group. In contrast, for L/D 8:12, *sjCRY-DASH* transcription decreased from ZT2 to ZT14 and remained steady from ZT14 to ZT22 (Figure 4C) with different response patterns.

### 2.4. Temperature Effects the Expression of sjCRY-DASH

The response of *sjCRY-DASH* to different temperatures was detected in juvenile sporophytes. At 5 °C, the initial expression level of *sjCRY-DASH* was higher, and no significant differences were observed along with the duration of exposure (Figure 5). At 10 °C, the *sjCRY-DASH* transcription drastically increased from ZT2 to ZT6 and then decreased, and the second maximum appeared at ZT18 (Figure 5). At 15 °C, the transcriptional level was lower than that at other temperatures and then slightly declined with treatment time. Generally, compared with the variation at both low and high temperatures, the drastic *sjCRY-DASH* transcriptional response at 10 °C exhibited the circadian rhythm.

### 2.5. SjCRY-DASH Is Negatively Regulated by Novel-m3234-5p

For small RNA libraries from dark, BL, and WL, a total of 31,717,782, 27,078,786, and 22,552,669 raw reads were generated, respectively (Appendix A). The mapping ratios between clean reads and the reference genome were in the range of 69.27–72.91% (Appendix A). Raw reads from small RNA sequencing were filtered to remove low quality sequences, and the majority of the small RNAs were distributed at a read length of 20–22 nt (Appendix A). The length distribution exhibited a prominent peak at 21 nt, representing 61% of the small RNAs. Nucleotide preference distributions revealed that most of the miRNAs tended to start with a 5′-U (Appendix A), which was consistent with typical sequence patterns of miRNA. Following the global identification of miRNAs, a novel miRNA, *novel-m3234-5p*, was predicted to target *sjCRY-DASH* (Figure 6A). The precursor secondary structure predicted by RNA fold further confirmed the presence of this candidate miRNA (Figure 6B). The complementarities between *sjCRY-DASH* and mature miRNA *novel-m3234-5p* sequences are marked in Figure 6A. qRT-PCR revealed that *novel-m3234-5p* transcription was clearly downregulated (Figure 6C) and that *sjCRY-DASH* transcription was upregulated by BL and WL compared with RL (Figure 3), suggesting that there was a negative correlation between *novel-m3234-5p* and *sjCRY-DASH*. Both *novel-m3234-5p* and *sjCRY-DASH* were detected in different tissues of *S. japonica*. Moreover, *novel-m3234-5p* displayed higher expression in stipes than in holdfasts and blades, whereas its target gene exhibited adverse patterns (Figure 7).

### 2.6. LincRNAs Regulate sjCRY-DASH via Trans-Acting mechanisms

The expression profile of lncRNA was analyzed by Illumina (paired-end) sequencing techniques. A total of 179,904,468, 190,361,256, and 192,670,442 raw reads were generated in the dark, BL, and WL libraries, respectively (Appendix A). Approximately 87.96%, 92.28%, and 91.47% clean reads in Dark, BL, and WL libraries were respectively mapped to the *S. japonica* reference genome (Appendix A). According to the location of identified lncRNAs relative to protein coding gens, 5206 lncRNAs were classified as long intergenic non-coding RNAs (lincRNAs), and 2540 lncRNAs were classified as natural sense lncRNAs (Appendix A). LncRNAs were able to regulate the gene expression by *cis* or *trans* action. The nearby gene that is less than 100 kb upstream or downstream away from the lncRNA can be regarded as the *cis*-acting target of lncRNA. However, no lncRNAs were verified for targeting *sjCRY-DASH* by *cis*-regulation manner. The *trans*-acting analysis found that the Pearson correlation coefficients between *sjCRY-DASH* and nine lncRNAs were above 0.99 (Table 1). These nine lncRNAs can be classified as lincRNAs and their coding capacities were identified as negative (Figure 8). qRT-PCR verification indicated that TCONS_00043396, TCONS_00043393, TCONS_00009907, TCONS_00008371, TCONS_00008286, and TCONS_00002718 were upregulated (Figure 6C) and were positively correlated with *sjCRY-DASH* transcription. Finally, a miRNA*-sjCRY-DASH*-lincRNA network was constructed (Figure 9).

### 2.7. Identification of the Candidate Competing Endogenous RNAs of sjCRY-DASH

To establish a *sjCRY*-*DASH*-miRNA*-*mRNA network (ceRNA network), candidate targets of *novel-m3234-5p* were searched based on the prediction result of PatMatch software. It was shown that the 25 targeted genes contain a sequence with almost complete complementarity to that of *novel-m3234-5p*, which were regarded as candidate competing endogenous RNAs for *sjCRY-DASH* (Table 2). Moreover, clustering analysis revealed that 14 of the 25 targeted genes exhibited similar expression patterns as *sjCRY*-*DASH* (Figure 10), indicating a possible negative regulation relationships with *novel-m3234-5p*. Following functional annotation, these 14 target genes were found to encode proteins involved in DNA damage repair, transcriptional regulation, carbohydrate metabolism, protein synthesis and processing, and actin transport (Figure 11).

## 3. Discussion

In the present study, we investigated the molecular structure and function of *CRY-DASH* in the typical photosynthetic stramenopile of *S. japonica*. Gene structural analysis showed that the length of the sjCRY-DASH ORF was 1779 bp, which could encode 592 amino acids. A single band was presented at ~66 kDa following heterologous expression, which was in accordance with the expected size of the gene product and close to the maximum size of cryptochrome family members (70 kDa) [17]. However, a distant relationship between *S. japonica* and other photosynthetic organisms in phylogeny analysis of CRY-DASH revealed a distinctiveness in evolution that might be associated with its adaptation to the local light regime [2].

The maximum levels of *CRY-DASH* transcription vary in plants in response to circadian rhythm, which might be attributed to varying light intensities and durations of local environments [27]. Following the induction of photoperiods, different patterns of maximum *sjCRY-DASH* transcription were observed, but the maxima occurred in 24 h cycles. This result indicated that *sjCRY-DASH* is under the control of circadian rhythm. Similar transcriptional patterns were observed under 10 °C, which implied that the circadian rhythm was constant in the suitable survival temperature of *S. japonica*. Once the temperature exceeded the tolerance of *S. japonica*, the transcription maxima were completely absent, suggesting that high seawater temperatures disturbed its circadian rhythm. Such self-maintaining circadian rhythms in different photoperiods and temperatures may allow sporophytes to cope with daily light and temperature regime changes and use the periodic properties to maximize the fitness when growing in coastal environments.

Circadian rhythm robustness depends on multiple feedback loops, and light is the most important environmental cue to entrain the circadian clock [28,29]. As the input circadian signals, BL and WL rapidly increased s*jCRY-DASH* transcription within 30 min, but the transcription did not change under RL. These results implied that BL could upregulate *sjCRY-DASH* transcription. Compared with BL-induced transcription, WL-induced transcription increased more slowly, and this discrepancy might be related to the influence of multiple wavelengths on the intricate regulation of *sjCRY-DASH* transcription. Additionally, WL contains blue light, and the slow induction effect of *sjCRY-DASH* might be attributed to a lower amount of BL in WL than that in the pure BL provided.

In addition to external environmental factors, the expression level of *sjCRY-DASH* was also regulated by endogenous non-coding RNAs. Our results showed that *novel-m3234-5p* could target *sjCRY-DASH*, and its binding sites in *sjCRY-DASH* are mainly mediated by a miRNA-induced silencing complex that could be destabilized by translating ribosomes [30]. Additionally, *novel-m3234-5p* significantly decreased within 3 h under BL, but transcription of its target gene increased, which indicated that the transcription of *novel-m3234-5p* was negatively correlated with that of *sjCRY-DASH*. In addition, *novel-m3234-5p* regulated *sjCRY-DASH* transcription by triggering its target sequence cleavage or degradation. Here, the decrease in *novel-m3234-5p* under BL might act as an internal negative modulator for promoting the accumulation of its target gene. In addition to microRNAs, six lncRNAs classified as lincRNAs were identified to act in *trans* with distal *sjCRY-DASH*. Previously, lincRNAs were regarded as regulators of chromatin remodeling during plant development [31]. Here, the lincRNAs were highly associated with the whole lncRNAs and accounted for 45% of all lncRNAs in *S. japonica*, indicating that lincRNAs are predominant in multi-cellular brown algae [25]. Moreover, there were co-expression patterns of lincRNAs and *sjCRY-DASH*, thus forming a miRNA-*sjCRY-DASH*-lincRNA network.

To further explore the possible function of *sjCRY-DASH*, we characterized its competing endogenous RNA (ceRNA) network in *S. japonica* following the identification of the miRNA-*sjCRY-DASH*-lincRNA network. Identification of ceRNA networks has been widely used in the research of human disease research to explore gene function, but studies of competing relationships in plants are still rare [32]. CeRNA clusters with similar expression patterns as *sjCRY-DASH* were identified in the present study. Functional analysis revealed that this cluster was mainly related to DNA damage repair, transcriptional regulation, carbohydrate metabolism, protein synthesis and processing, and actin transport. This indicated that *sjCRY-DASH* might function in the regulation of numerous biological processes, which is consistent with the contrasting functions of *CRY-DASH* in animals, plants, and fungi [14,33,34]. Due to the technical barrier in the construction of transgenetic macroalgae, the function of *sjCRY-DASH* is difficult to further identify. Nonetheless, our results provide the potential direction for the study of CRY-DASH function.

In conclusion, *sjCRY-DASH* transcription is regulated by circadian rhythms with 24 h periodicity, as shown by different photoperiods across a suitable temperature. BL can rapidly upregulate *sjCRY-DASH* transcription via the downregulation of *novel-m3234-5p*, and *sjCRY-DASH* is concurrently co-expressed with six lincRNAs; therefore, these non-coding RNAs play an important role in the regulation of *S. japonica* circadian rhythms during physiological adaptation.

## 4. Materials and Methods

### 4.1. Sample Collection and Treatment with Various Conditions

*S. japonica* “Zhong ke No. 2” juvenile sporophytes (15–35 cm) were collected from the cultivation rafts of an experimental field in Rongcheng, Shandong, China, in 2016. Following collection, sporophytes were carefully washed with filtered sterile seawater several times to eliminate the epiphytes and sediment, and then were precultured in darkness at 10 °C overnight. The juvenile sporophytes were treated under various illumination conditions, first in the dark (60 h) and then exposed to BL (25 µmol m^−2^ s^−1^), WL (70 µmol m^−2^ s^−1^), or RL (40 µmol m^−2^ s^−1^) for 5 h. For the photoperiod treatments, the sporophytes were cultured under L/D 16:8, 12:12, or 8:16 L:D cycles for 72 h. After 72 h, they were collected at 4 h intervals. In addition, the juvenile sporophytes were cultured at 5 °C, 10 °C, and 15 °C under L/D 12:12 and collected at 4 h intervals. Each treatment was conducted with six biological replicates. After the treatment, all kelp samples were rapidly frozen in liquid nitrogen and were then preserved at −80 °C.

### 4.2. RNA Preparation and cDNA Synthesis

Total RNAs from each frozen sample were extracted with a plant RNA kit (Omega Bio-Tek, Norcross, GA, USA) according to the manufacturer’s instructions. The quality and concentration of the RNA product was examined by both agarose gel electrophoresis and a DS-11 spectrophotometer (DeNovix, Wilmington, DE, USA). First-strand cDNA was subsequently reverse transcribed from the high-purity RNA (the ratio of optical density at 260 and 280 nm, OD260/280 = 1.8–2.2) according to the protocol of the PrimeScript II 1st-strand cDNA synthesis kit (TaKaRa, Dalian, China).

### 4.3. Cloning of CRY-DASH from S. japonica

The *sjCRY-DASH* unigene was retrieved from the transcriptome database of *S. japonica* (GSE33853) with the BLASTX tool [35]. Based on the retrieved fragment of the *sjCRY-DASH* transcript, RACE was performed to clone the full-length of *sjCRY-DASH* according to the protocols of both the SMARTer RACE cDNA amplification kit (Clontech, Mountain View, CA, USA), and 3′-Full RACE Core Set Ver.2.0 (TaKaRa, Dalian, China). Trans-*Taq* High Fidelity PCR SuperMix II (TransGen Biotech, Beijing, China) was used, and PCR was performed under the following conditions: 94 °C for 5 min; 30 cycles of 94 °C for 30 s, 56 °C for 30 s, and 72 °C for 2 min; and 72 °C for 10 min. The PCR products were ligated into the pGEM-T Easy vector (Promega, Madison, WI, USA), and clones containing approximately 1000 bp and 750 bp fragments were identified. Following DNA sequencing, DNAMAN software (Version 6.0, Lynnon Biosoft, Quebec, Canada) was used to align the sequence. All the specific primers are listed in Appendix A.

### 4.4. Heterologous Expression and Purification of sjCRY-DASH

A cold shock expression system (TaKaRa, Dalian, China) was used to analyze the expression of the His-tagged protein. Specific primers with *Nde*I and *EcoR*I digestion sites (CRY-DASH-F and CRY-DASH-R) were designed to amplify the ORF of *sjCRY-DASH*. The purified amplification products were ligated to the pMD-19T vector (TaKaRa, Tokyo, Japan) and then digested with *Nde*I and *EcoR*I. Target bands were selected, purified, and recombined into the pCold I vector. The His-tagged protein was expressed in *E. coli* Transetta DE3 (Transgene, Beijing, China) in the LB medium with 50 mg/L ampicillin. We first incubated the *E. coli* in 10 mL medium, and after the OD_600_ reached 0.6, we added 0.5 mM isopropyl-β-D-thiogalactoside for continuous culture overnight at 15 °C. The medium was harvested and then centrifuged at 3552× *g* for 10 min, after which the sediments were collected and resuspended in a new tube containing 500 µL PBS buffer (pH 7.4). After disrupting the cellular solution by sonication, 40 µL samples were used for SDS-PAGE (12%).

To purify the expressed sjCRY-DASH, after centrifugation the supernatant was removed and applied to a His affinity column (GE Healthcare, Piscataway, NJ, USA). Then, the column was washed with binding buffer (8 M urea; 50 mM Tris-HCl, pH 8.0; 300 mM NaCl; and 20 mM imidazole), which was followed by elution with 500 mM of imidazole. Finally, crude protein was collected, dialyzed in PBS at pH 7.4, and stored at −80 °C.

### 4.5. Small RNA Library Preparation and Sequencing

After exposure to dark, BL, or WL for 3 h, total RNA was extracted from juvenile sporophytes and treated with TRIzol reagent (Invitrogen, Carlsbad, CA, USA). After assessing the quality of the RNA with an Agilent 2100 bioanalyzer system (Agilent Technologies, Palo Alto, CA, USA), the RNAs with high purity (OD_260/280_ = 1.8–2.2) and integrity (RNA integrity number, RIN > 7.5) were adjusted to equivalent concentrations to construct the small RNA libraries. Small RNAs of 18–30 nt in length were first isolated by size using PAGE, then ligated with 3′ and 5′ adapters to amplify cDNA by reverse transcription PCR, and finally sequenced by an Illumina HiSeq 2500. The raw data obtained from sequencing were filtered to remove low quality sequences and the obtained clean reads were compared with our *S. Japonica* genome data (GenBank accession number: MEHQ00000000) using searchers in the Rfam (http://rfam.sanger.ac.uk/) and miRBase (http://www.mirbase.org/) databases to identify miRNAs. Unidentified reads with candidate precursors were predicted as novel miRNAs via MIREAP software (https://sourceforge.net/projects/mireap/).

### 4.6. Long Non-Coding RNA Library Preparation and Sequencing

Total RNA was re-purified to remove rRNA using the Epicentre Ribo-Zero rRNA removal kit (Epicentre, Madison, WI, USA), and broken into short fragments to serve as a template for first strand cDNA. Second strand cDNA was synthesized using RNase H (Invitrogen, Carlsbad, CA, USA) and DNA polymerase (Invitrogen, Carlsbad, CA, USA). cDNA fragments for the ligating adapters were purified using the QIAquick PCR purification kit (Qiagen, Hilden, Germany), and the constructed cDNA library was sequenced by an Illumina HiSeq 4000 (Illumina, San Diego, CA, USA). After the raw reads were filtered, clean reads were mapped to our reference genome of *S. japonica*. Cufflinks was used for reconstructing the transcript data and obtaining the known transcript profile with Coding-Non-Coding Index software to predict the coding capacity of the new transcript. Then, the lncRNA sequences were finally obtained.

### 4.7. miRNA-sjCRY-DASH-lncRNA Association Analysis

According to the identified miRNAs, the candidate target genes were predicted using PatMatch_v1.2 software (https://www.arabidopsis.org/cgi-bin/patmatch/nph-patmatch.pl) following conditions: (1) No more than four mismatches between the miRNA and target, where G-U base count as 0.5 mismatches; (2) no mismatches in positions 10–11 of the miRNA/target duplex; (3) no more than 2.5 mismatches in positions 1–12 of the 5′ end of the miRNA; and (4) the minimum free energy of the miRNA/target duplex was greater than or equal to 74% of the free energy of miRNA bound to its perfect complement. The proximal protein coding genes located within a genomic window of 100 kb were sought as the *cis*-regulators of lncRNAs [36]. Correlation of expression between the lncRNAs and protein coding genes was used to screen the target genes of the lncRNAs that were involved *in trans* regulation, and Pearson correlation coefficients with absolute values over 0.9 were required as the criteria. Finally, non-coding RNAs associated with the regulation of *sjCRY-DASH* were screened and used for miRNA-*sjCRY-DASH*-lncRNA network construction.

### 4.8. Quantitative Real-Time PCR Analysis

Total RNA (1 µg) was used for reverse transcription using the PrimeScript RT reagent kit with gDNA Eraser (TaKaRa, Tokyo, Japan). Real-time PCR was performed with SYBR Premix Ex Taq II (TaKaRa, Tokyo, Japan) on a TP800 Thermal Cycler Dice Real-Time System (TaKaRa, Japan). The cycling conditions included an initial incubation at 95 °C for 30 s, followed by 40 cycles of 95 °C for 5 s, and 60 °C for 30 s. The relative transcript abundances of the target genes were calculated based on the 2^−ΔΔCt^ method [37]. All of the experiments were performed in six biological replicates. Statistical differences in gene expression were examined using one-way or two-way analysis of variance (ANOVA) with SPSS 18.0 (SPSS Inc., Chicago, IL, USA). *P*-values lower than 0.05 were considered to be significant. Actin and U6 genes were used as internal controls. The primers used for qPCR amplification are listed in Appendix A.

## Figures and Tables

**Figure 1 ijms-21-00309-f001:**
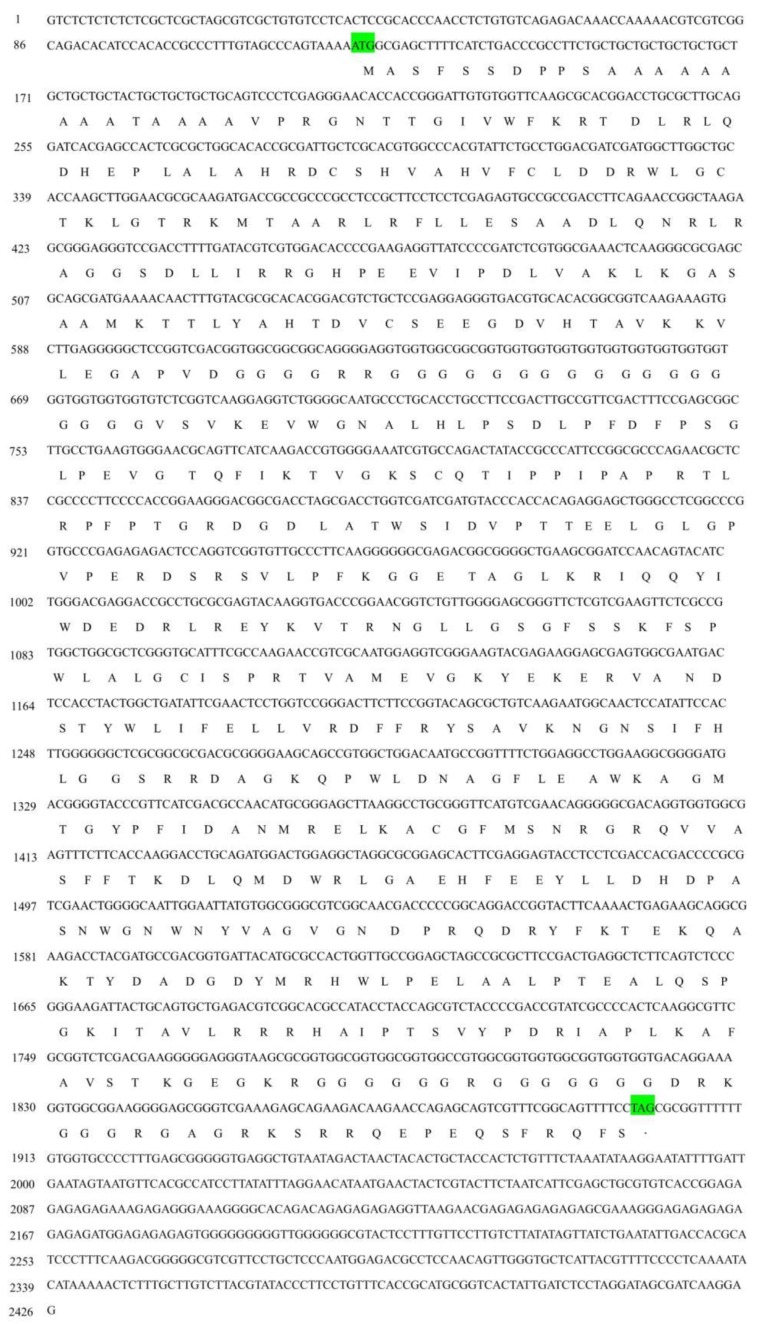
The full-length cDNA and the deduced amino acid sequences of sjCRY-DASH from *Saccharina japonica*. The letters in green indicate the start codon (ATG) and stop codon (TAG).

**Figure 2 ijms-21-00309-f002:**
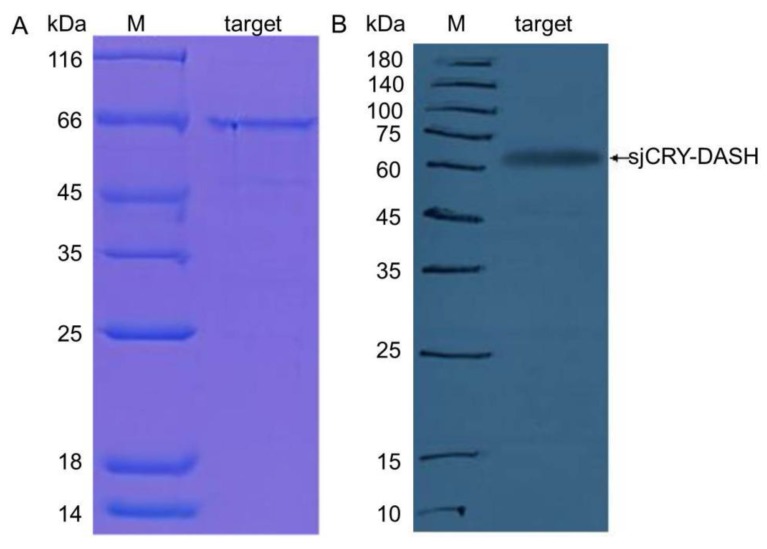
(**A**) Heterologous expression of sjCRY-DASH. SDS-PAGE analysis of purified sjCRY-DASH; M, protein marker; Lane target, eluted fraction with the presence of His-tagged sjCRY-DASH. (**B**) Western blot analysis of purified sjCRY-DASH. M, protein marker; target, eluted fraction with the presence of His-tagged SjCRY-DASH.

**Figure 3 ijms-21-00309-f003:**
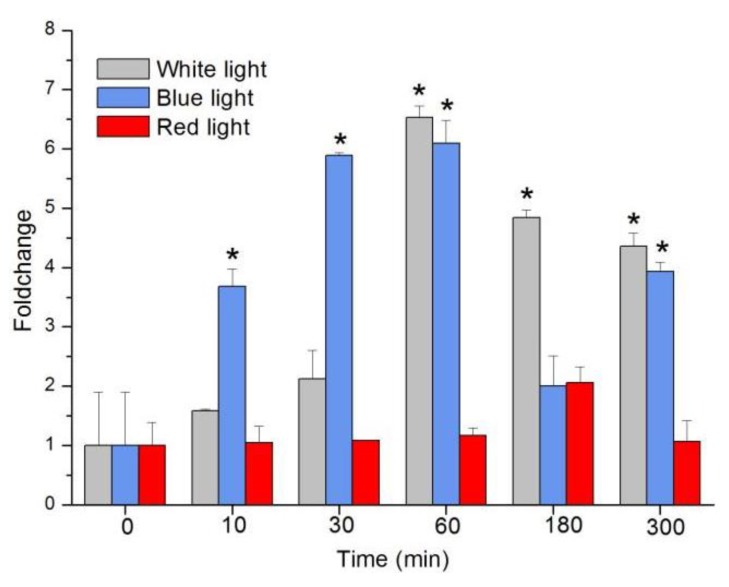
Transcription changes of *sjCRY-DASH* induced by white light, blue light, and red light. Transcription accumulation was quantified by qRT-PCR. The changes in transcript levels after exposure to different light conditions are presented as fold changes relative to the RNA from dark-grown sporophytes. Each test was performed in six biological samples. The data in the figures represent the averages ± standard deviation. Data were analyzed by two-way ANOVA followed by Turkey’s multiple comparison test. * *p* < 0.05.

**Figure 4 ijms-21-00309-f004:**
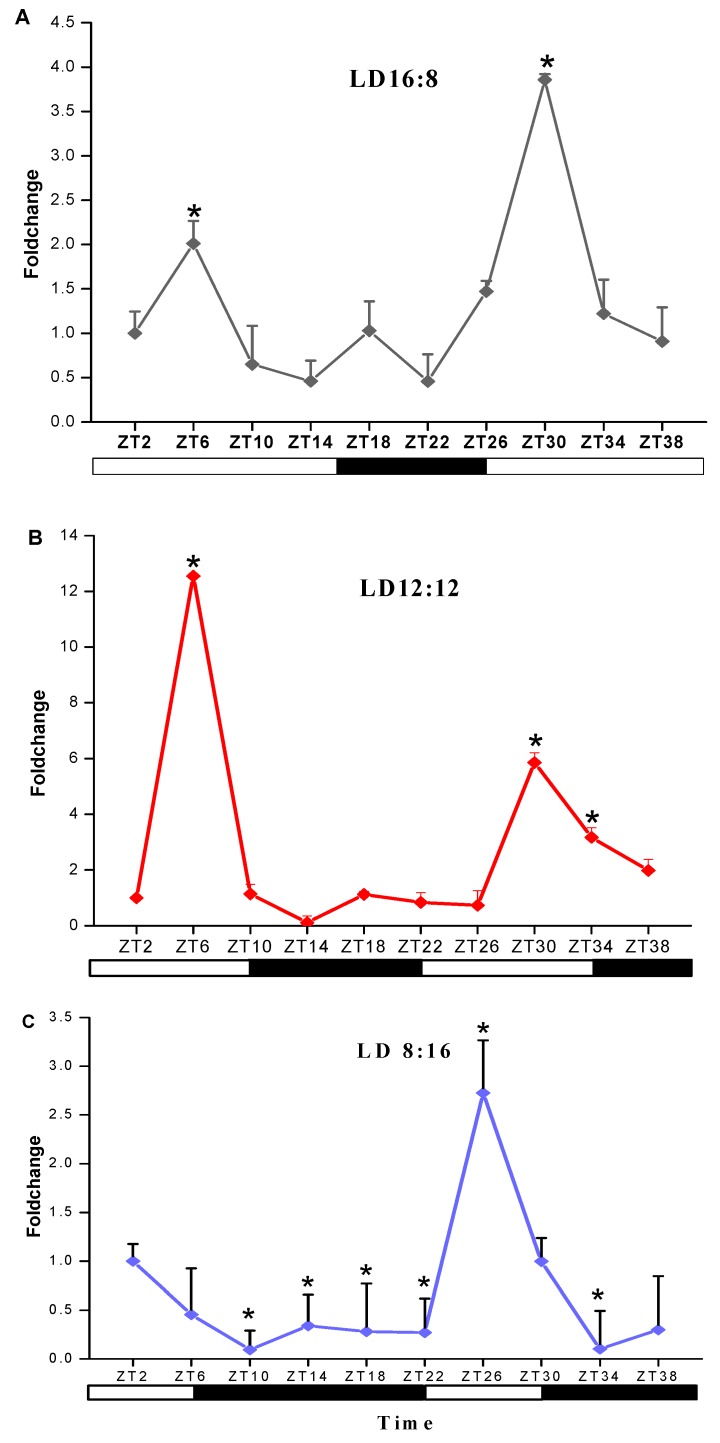
Transcription analysis of *sjCRY-DASH* under light/dark (LD) treatments of (**A**) LD16:8, (**B**) LD12:12 and (**C**) LD8:16. The expression levels at different times are presented as the fold change relative to that at Zeitgeber Time 2 (ZT2). Each test was performed in six biological samples. The data in the figures represent the averages ± standard deviation.* *p* < 0.05 based on one-way ANOVA.

**Figure 5 ijms-21-00309-f005:**
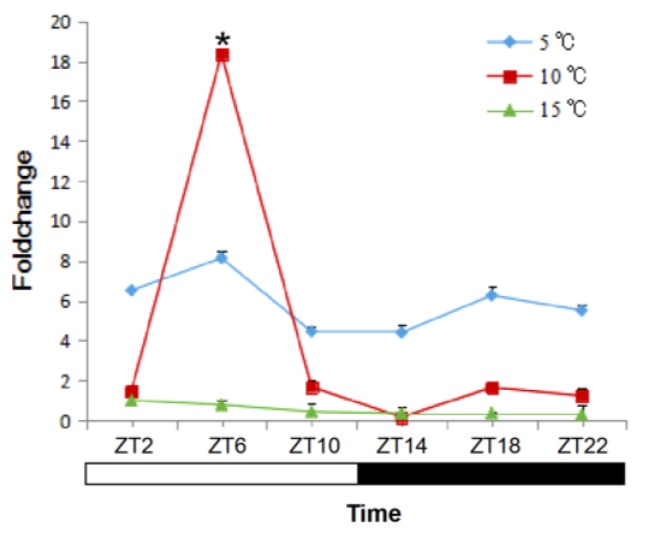
Quantification of *sjCRY-DASH* transcriptional levels under different temperatures. Juvenile sporophytes of *S. japonica* were grown under the light/dark (LD) 12:12 cycle and harvested at 4 h intervals. The expression levels at the treatment times are presented as the fold change relative to that at ZT2. Each test was performed in six biological samples. The data in the figures represent the averages ± standard deviation. * *p* < 0.05 based on one-way ANOVA.

**Figure 6 ijms-21-00309-f006:**
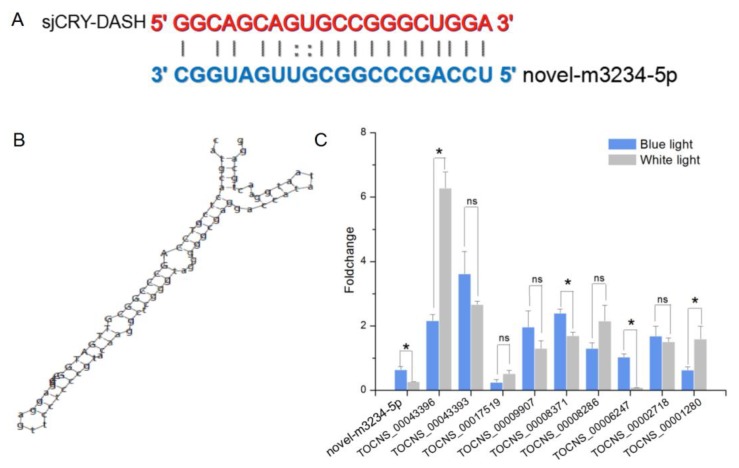
Identification of *sjCRY-DASH* as a target gene of novel-m3234 in *S. japonica*. (**A**) Base-pairing interaction between *novel-m3234-5p* and *sjCRY-DASH*. (**B**) Secondary structure of the predator sequence of *novel-m3234-5p*. (**C**) qRT-PCR analysis of miRNA and lncRNA expression that targeted *sjCRY-DASH* genes. Non-coding RNA expression levels in the dark were used for data normalization. Each value was performed in six biological samples. The data in the figures represent the averages ± standard deviation. One-way ANOVA was used for statistical comparisons between groups. * *p* < 0.05, and ns = not significant.

**Figure 7 ijms-21-00309-f007:**
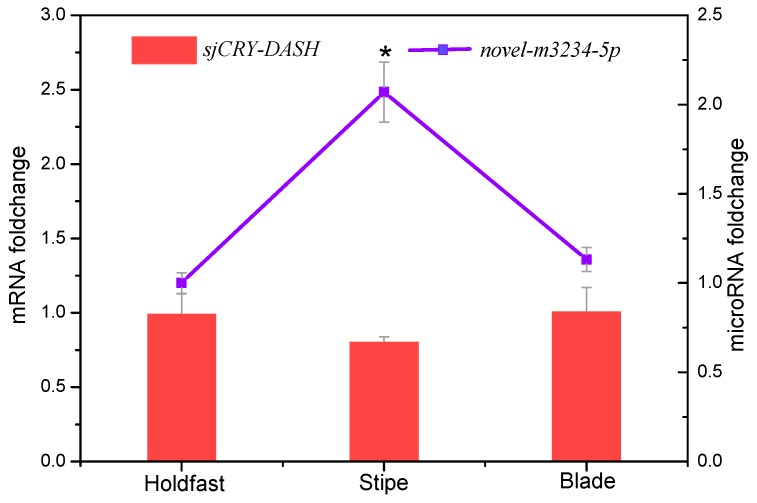
Tissue-specific transcription expression of *novel-m3234-5p* (point) and *sjCRY-DASH* (bar) in *S. japonica.* Each value was performed in six biological samples. The data in the figures represent the averages ± standard deviation. * *p* < 0.05 based on one-way ANOVA.

**Figure 8 ijms-21-00309-f008:**
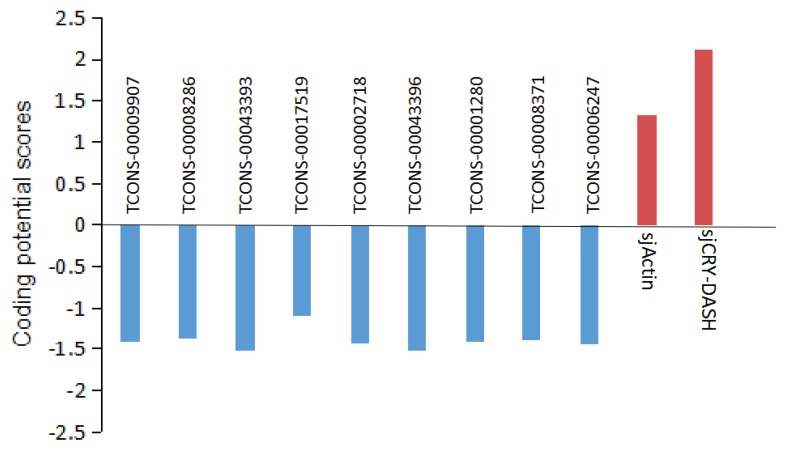
Analysis of the coding potential of lncRNA. Coding potential scores were generated using the Coding Potential Calculator (CPC) program. Transcripts with scores beyond −1 and 1 are marked as non-coding and coding in the CPC classification, respectively. *sjActin* and *sjCRY-DASH* are provided as coding examples.

**Figure 9 ijms-21-00309-f009:**
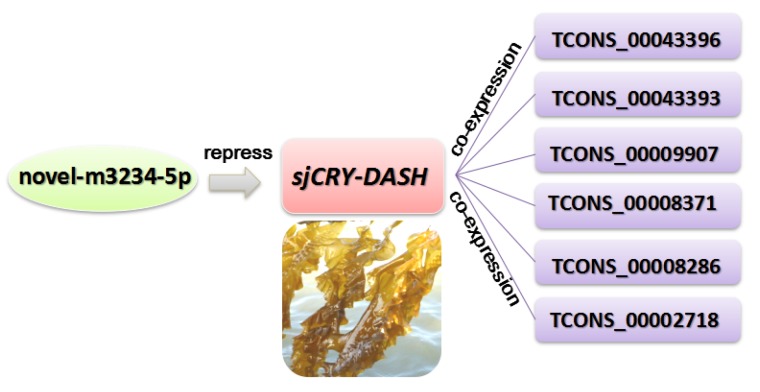
A miRNA*-sjCRY-DASH*-lincRNA network in *S. japonica*.

**Figure 10 ijms-21-00309-f010:**
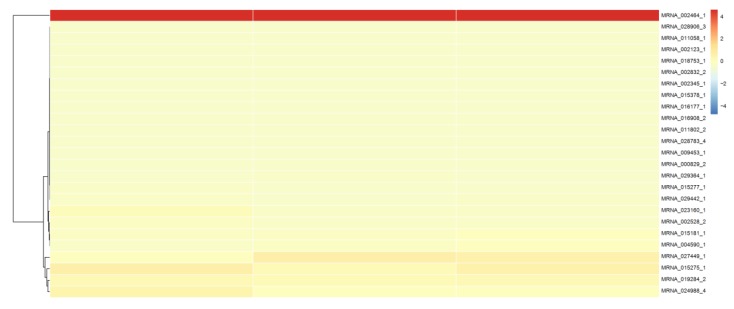
Cluster analysis of the transcript expression pattern of target RNAs of *novel-m3234-5p* in *S. japonica*.

**Figure 11 ijms-21-00309-f011:**
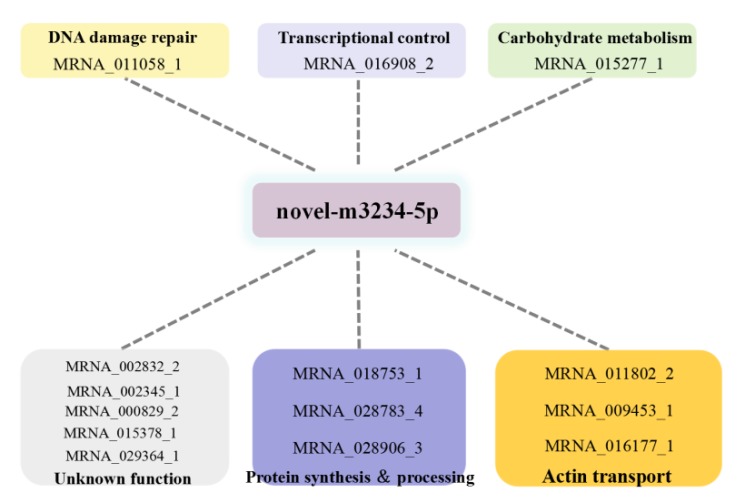
Functional annotation of competing endogenous RNAs of *sjCRY-DASH* that were targeted to *novel-m3234-5p*.

**Table 1 ijms-21-00309-t001:** *Trans*-action of lncRNA.

lncRNA_id	Classification	Target	Correlation
TCONS_00043396	LincRNA	*SjCRY-DASH*	0.999986
TCONS_00017519	LincRNA	*SjCRY-DASH*	0.999975
TCONS_00009907	LincRNA	*SjCRY-DASH*	0.999995
TCONS_00008371	LincRNA	*SjCRY-DASH*	0.999993
TCONS_00008286	LincRNA	*SjCRY-DASH*	0.999996
TCONS_00006247	LincRNA	*SjCRY-DASH*	0.999996
TCONS_00002718	LincRNA	*SjCRY-DASH*	0.999996
TCONS_00001280	LincRNA	*SjCRY-DASH*	0.999998
TCONS_00043393	LincRNA	*SjCRY-DASH*	0.99999

**Table 2 ijms-21-00309-t002:** Complementary between *novel-m3234-5p* and mRNAs.

microRNA	Target	Target_Region (3′ → 5′)	Pairing (5′ → 3′)	miRNA_Sequence (5′ → 3′)
novel-m3234-5p	MRNA_015181_1	AGGUGGGGCCGGAAGUACCU	||||x||||||x||x||||x	UCCAGCCCGGCGUUGAUGGC
novel-m3234-5p	MRNA_002832_2	AGGUACGGUCGCAACAACCG	||||xx||o||||||x||||	UCCAGCCCGGCGUUGAUGGC
novel-m3234-5p	MRNA_011802_2	AGGGCGGGUCGAAACUACUG	|||x||||o||x||||||o|	UCCAGCCCGGCGUUGAUGGC
novel-m3234-5p	MRNA_016177_1	AGGUCGAGCCGCUGCUACAG	||||||x|||||xo||||x|	UCCAGCCCGGCGUUGAUGGC
novel-m3234-5p	MRNA_009453_1	GGGCGGGGCCGCAGCUACCG	o||xx||||||||o||||||	UCCAGCCCGGCGUUGAUGGC
novel-m3234-5p	MRNA_016908_2	AGGUCGUGCCGCAACAACGA	||||||x||||||||x||xx	UCCAGCCCGGCGUUGAUGGC
novel-m3234-5p	MRNA_015275_1	AGGUCGGGCCGGAACGACGA	|||||||||||x|||x||xx	UCCAGCCCGGCGUUGAUGGC
novel-m3234-5p	MRNA_015277_1	GGGCGGGGCCGCAGCUACCG	o||xx||||||||o||||||	UCCAGCCCGGCGUUGAUGGC
novel-m3234-5p	MRNA_004590_1	AGCUUGUGCCGCAACUACUG	||x|o|x|||||||||||o|	UCCAGCCCGGCGUUGAUGGC
novel-m3234-5p	MRNA_000829_2	AGGCAGGGCCGCAACAACCC	|||xx||||||||||x|||x	UCCAGCCCGGCGUUGAUGGC
novel-m3234-5p	MRNA_023160_1	CGGUCGGGCCGCAACUGCUA	x|||||||||||||||o|ox	UCCAGCCCGGCGUUGAUGGC
novel-m3234-5p	MRNA_019284_2	AGGACAGGCCGCAACUUCCA	|||x|x||||||||||x||x	UCCAGCCCGGCGUUGAUGGC
novel-m3234-5p	MRNA_002345_1	AGGUCGAGCCGCAACAGCCG	||||||x||||||||xo|||	UCCAGCCCGGCGUUGAUGGC
novel-m3234-5p	MRNA_011058_1	AGGUGGGGACGCAGGUACCG	||||x|||x||||ox|||||	UCCAGCCCGGCGUUGAUGGC
novel-m3234-5p	MRNA_029442_1	GGGUCGUGCCGCAACUUACG	o|||||x|||||||||xx||	UCCAGCCCGGCGUUGAUGGC
novel-m3234-5p	MRNA_027449_1	AGGUGGGGCCGCUACCGCCG	||||x|||||||x||xo|||	UCCAGCCCGGCGUUGAUGGC
novel-m3234-5p	MRNA_024988_4	AGGGAGGGCCGCAACUUCCC	|||xx|||||||||||x||x	UCCAGCCCGGCGUUGAUGGC
novel-m3234-5p	MRNA_018753_1	GGGUCCGGCCGCAUCUACCC	o||||x|||||||x|||||x	UCCAGCCCGGCGUUGAUGGC
novel-m3234-5p	MRNA_002528_2	AGGACGCGCCGCAACUGCCG	|||x||x|||||||||o|||	UCCAGCCCGGCGUUGAUGGC
novel-m3234-5p	MRNA_002464_1	AGGACAGGCCGCGACUACCA	|||x|x||||||o||||||x	UCCAGCCCGGCGUUGAUGGC
novel-m3234-5p	MRNA_029364_1	AGGUGGGGCCGAAGCUAGCG	||||x||||||x|o|||x||	UCCAGCCCGGCGUUGAUGGC
novel-m3234-5p	MRNA_028783_4	AGGUCGGGUCGCAACUACGA	||||||||o|||||||||xx	UCCAGCCCGGCGUUGAUGGC
novel-m3234-5p	MRNA_015378_1	AGGUCGAGCCGUAACUAGUG	||||||x||||o|||||xo|	UCCAGCCCGGCGUUGAUGGC
novel-m3234-5p	MRNA_028906_3	AGCUAGGGCCGCUGCUACCG	||x|x|||||||xo||||||	UCCAGCCCGGCGUUGAUGGC

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
