# Peer review of "Non-Coding RNAs Participate in the Regulation of CRY-DASH in the Growth and Early Development of Saccharina japonica (Laminariales, Phaeophyceae)"

_ijms, 2020, doi:10.3390/ijms21010309_

Round 1
Reviewer 1 Report
Dear authors,
you investigated the possible role of non- coding RNAs in Saccharina japonica in early development and growth.The authors performed exposure experiments, smallRNA & long non-coding RNA sequencing, as well as validation of RNAseq data by qPCR.
I acknowledge that this study represents considerable work. However, I have some concerns regarding some methodological aspects. Even though the topic is new in brown algae and quite interesting, the research is not adequately described, especially in the material & methods section. Here some pinpoints:
Introduction: Would benefit from English language editing
Material & methods:
For the used material: You do not mention possible contaminations of the material - the used sporophytes seem to origin from cultivation rafts. Was the material examined somehow? What about contamination with diatoms?
Experimental set-up:You did not describe (for most of the experiments and methods) how many technical and biological were used. Only for the qPCR it is mentioned that 3 technical and 2 biological replicates were used, which is in my opinion not technically sound, one should at least use 3 biological replicates.
Analysis: Statistical analysis of the data has not been in the appropriate way.
Results: Mapping rates of the RNA seq data on the Saccharina latissima genome is missing. Are there any decontamination procedures to eliminate contaminant sequences from the raw sequencing data, such as bacteria or diatom sequences?
I would suggest a major revision of the manuscript as well as a repetition of the qPCR with at least 3 biological replicates.
Best regards
Reviewer 2 Report
The authors identfied and characterized expression of a Cry-Dash protein from S. japonica under different light conditions and temperatures. They could identify non-coding RNAs that either seem to act as a repressor or are co-transcribed with Cry-Dash. Thus, this study sheds light on the regulation of DNA damage repair and/or photoperception in this species by non-coding RNAs.
major comments:
It is unfortunate, that the purified Cry-DASH has not been used for any experiments. It would be interesting, for example, to see whether it has DNA repair activity or only acts as a photoreceptor (using an electrophoretic mobility shift assay (EMSA)), although it might be beyond the scope of this paper.
Figure 3: Figure 3A and B should be merged into a single diagram, to make this lack of regulation under red light more obvious. Furthermore, timepoints 180 and 300min are ignored in the main text.
Figure 5: In my opinion, only the 10°C sample behaves as shown previously in Figure 4. Neither the 5°C nor the 15°C sample show a circadian rhythm, the only difference between them is that the 5°C sample has a higher inital expression level, but no significant differences were observed for the different timepoints. Thus, not only the 15°C sample (as mentioned in the discussion) but also the 5°C sample seems to be deregulated.
Figure 6: What are the white bars and why don't all samples have one? It's not mentioned in the figure legend.
minor comments:
Line 86, 254: Expression in E. coli is not in vitro expression, it's heterologous expression
line 90: As a purified protein is used for Western Blot, it cannot be said that the antibody binds with high specificity. There is nothing else to bind to if the same fraction as in Figure 2A is used.
line 126: spelling error: It is Tukey's test
line 186: it's unclear what "its expression" refers to, presumably Cry-Dash?
line 210-215: Why were the other RNAs chosen? Similar expression? This is unclear from the results chapter.
Line 275: As the white light spectrum might also contain blue light, the reduced induction might be due to a lower amount of blue light compared to the blue light condition
Figures 3/4/5: Significance threshold should be mentioned in the text and in the figure legend.
Figure 4: Description of replicates is unclear. If expression levels are the mean of 3 independent experiments, why are there only at least two different biological samples?
Author Response
Dear Reviewer,
Thank you for your comments concerning our manuscript entitled “Non-coding RNAs participate in the regulation of CRY-DASH in growth and early development of Saccharina japonica (Laminariales, Phaeophyceae)” (ijms-612197). Those comments are all valuable and very helpful for revising and improving our paper, as well as the important guiding significance to our researches. We have studied comments carefully and have made extensive correction which we hope meet with approval. The responds to the reviewer’s comments are as follows:
1 It is unfortunate, that the purified Cry-DASH has not been used for any experiments. It would be interesting, for example, to see whether it has DNA repair activity or only acts as a photoreceptor (using an electrophoretic mobility shift assay (EMSA)), although it might be beyond the scope of this paper.
Response: Thank you for your suggestion. We agreed that it would be interesting to further determine the role of purified sjCRY-DASH played in DNA reparation and blue light reception. However, the main purpose of this paper is to explore the regulation mechanism of sjCRY-DASH in S. japonica. So the function identification of purified sjCRY-DASH was not added in this paper. We appreciate that you provide a good idea for us to further identify the function of purified sjCRY-DASH using electrophoretic mobility shift assay (EMSA) in the future work.
2 Figure 3: Figure 3A and B should be merged into a single diagram, to make this lack of regulation under red light more obvious. Furthermore, timepoints 180 and 300min are ignored in the main text.
Response: Thank you for your suggestion. We agreed that the merging of Fig. 3A and 3B could make the lack of regulation under red light more obvious. And we have merged these two figures into one figure in the revised manuscript. Additionally, the description of timepoints 180 and 300min also has been added in the revised manuscript.
3 Figure 5: In my opinion, only the 10°C sample behaves as shown previously in Figure 4. Neither the 5°C nor the 15°C sample show a circadian rhythm, the only difference between them is that the 5°C sample has a higher inital expression level, but no significant differences were observed for the different timepoints. Thus, not only the 15°C sample (as mentioned in the discussion) but also the 5°C sample seems to be deregulated.
Response: Thank you for your suggestion. We agreed that the sjCRY-DASH transcriptions at 5°C had no significant difference in different time points. Following careful consideration, we corrected this point in the discussion section in the revised manuscript.
4 Figure 6: What are the white bars and why don't all samples have one? It's not mentioned in the figure legend.
Response: Thank you for your comment. The white bars that we wanted to exhibit is the comparison between two means. Initially, we just labeled the samples with significant difference. According to reviewer’s suggestion, we added white bars to the all samples including those samples with no significant difference. And the corresponding explanation has been added in the revised manuscript.
5 Line 86, 254: Expression in E. coli is not in vitro expression, it's heterologous expression.
Response: Thanks a lot for reviewer’s comment. We have replaced “in vitro expression” as “heterologous expression” in the revised manuscript.
6 line 90: As a purified protein is used for Western Blot, it cannot be said that the antibody binds with high specificity. There is nothing else to bind to if the same fraction as in Figure 2A is used.
Response: Yes, we agree with reviewer’s comments that the binding of purified protein to antibody cannot be considered as “high specificity”. And we have deleted the wrong expression of “with high specificity” in the revised manuscript.
7 line 126: spelling error: It is Tukey's test.
Response: Thank you for reviewer’s careful review. We have corrected the spelling error in the revised manuscript.
8 line 186: it's unclear what "its expression" refers to, presumably Cry-Dash?
Response: Yes, we did not interpret this statement clearly. Here, “its expression” refers to “sjCRY-DASH expression”. We have corrected this sentence in the revised manuscript.
9 line 210-215: Why were the other RNAs chosen? Similar expression? This is unclear from the results chapter.
Response: We did not interpret this statement clearly. lncRNAs are found to regulate gene expressions both on cis and trans manner, and the location of the target gene is commonly used to distinguish between the two (Li et al. 2017). Cis-acting lncRNAs regulate the expression of target genes that are located at or near the same genomic locus, whereas trans-acting lncRNAs can either inhibit or activate gene transcription at independent chromosomal loci (Fatica and Bozzoni, 2014). The nearby gene that is less than 100kb upstream or downstream away from the lncRNA can be the potential target regulated by that lncRNA in cis manner. Following search the analysis results, no lncRNAs were found to target sjCRY-DASH by cis manner. To classify lncRNA trans-target genes, the BLAST software was used to assess the impact of lncRNA binding on complete mRNA molecules; the RNAplex program was then used to identify possible trans-target genes of the lncRNAs, enriching those mRNAs co-expressed with lncRNAs. The co-expression analysis was based on calculating the Pearson correlation coefficient (PCC) between coding genes and non-coding transcripts according to their expression levels. Following search the co-expression pairs between lncRNA and mRNA, we found that nine lncRNAs (as listed in Table 2) can target to sjCRY-DASH by trans-action. Then we further measure these lncRNAs expression by qPCR and six out of nine lncRNAs (TCONS_00043396, TCONS_00043393, TCONS_00009907, TCONS_00008371, TCONS_00008286, and TCONS_00002718) were identified to be co-expressed with sjCRY-DASH. So we deduced the lncRNAs-sjCRY-DASH network.
10 Line 275: As the white light spectrum might also contain blue light, the reduced induction might be due to a lower amount of blue light compared to the blue light condition.
Response: We agree with the reviewer’s suggestion. The spectrum of white light contains all of the visible light with wavelengths between 400nm and 760nm (Osborne et al. 2008). Blue light is also included in white light spectrum. According to reviewer’s suggestion, we have added this possibility in the discussion section of revised manuscript.
11 Figures 3/4/5: Significance threshold should be mentioned in the text and in the figure legend.
Response: Thank you for your suggestion. The significance threshold of p<0.05 has been added in the text and figure legend in the revised manuscript.
12 Figure 4: Description of replicates is unclear. If expression levels are the mean of 3 independent experiments, why are there only at least two different biological samples?
Response: Thank you for your comment. We did not interpret this statement clearly. The meaning of sentence “Each experiment was performed in triplicate with at least two different biological samples.” that we initially wanted to express was that every treatment was repeated for three times and every times has two biological samples. So every data in the qRT-PCR was the average values of six biological samples. Considering the inappropriate expression, we have corrected this sentence in the revised manuscript. The clear description of replicates has been added in the “Materials and Methods” and “Figure legends”.
Reference:
Osborne, N. N., Li, G. Y., Ji, D., Mortiboys, H. J., & Jackson, S. (2008). Light affects mitochondria to cause apoptosis to cultured cells: possible relevance to ganglion cell death in certain optic neuropathies. Journal of neurochemistry, 105(5), 2013-2028.
Fatica, A., & Bozzoni, I. (2014). Long non-coding RNAs: new players in cell differentiation and development. Nature Reviews Genetics, 15(1), 7.
Li, Z., Ouyang, H., Zheng, M., Cai, B., Han, P., Abdalla, B. A., ... & Zhang, X. (2017). Integrated analysis of long non-coding RNAs (LncRNAs) and mRNA expression profiles reveals the potential role of LncRNAs in skeletal muscle development of the chicken. Frontiers in physiology, 7, 687.
We tried our best to improve the manuscript and made extensive revision in the manuscript. We appreciate for Editors/Reviewers’ warm work earnestly, and hope that the correction will meet with approval.
Thank you and best regards.
Yours sincerely,
Xiaoqi Yang